# VideoShield: A Unified Framework for Multimodal Risk Detection and Control in Video Generative Models

## Abstract

Recent progress in video generative models has enabled the creation of high-quality videos from multimodal prompts that combine text and images. While these systems offer enhanced controllability and creative potential, they also introduce new safety risks, as harmful content can emerge not only from individual modalities but also from their interaction. Existing safety methods, primarily designed for unimodal settings, struggle to handle such compositional risks. To address this challenge, we present VideoShield, a unified safeguard framework for proactively detecting and mitigating unsafe semantics in multimodal video generation. VideoShield operates in two stages: First, a contrastive detection module identifies latent safety risks by projecting fused image-text inputs into a structured concept space; Second, a semantic suppression mechanism intervenes in the embedding space to remove unsafe concepts during generation. To support this framework, we introduce ConceptRisk, a large-scale, concept-centric dataset that captures a wide range of multimodal safety scenarios, including single-modality, compositional, and adversarial risks. Comprehensive experiments under various settings show that VideoShield consistently outperforms existing baselines, achieving state-of-the-art results in both risk detection and safe video generation.

## 1 Introduction

Recent advances in video generative models have enabled the synthesis of realistic and coherent videos from natural language prompts, visual references, or their combination. These capabilities are driven by large-scale diffusion models and multimodal learning architectures, which have significantly enhanced the quality and controllability of generated content (Ho et al., 2022; Singer et al., 2022; Khachatryan et al., 2023; Jiang et al., 2024b; Brooks et al., 2024; Bar-Tal et al., 2024). As such systems are increasingly applied in creative, educational, and simulation contexts, concerns over their safety have become more pressing(Miao et al., 2024; Ying et al., 2025; Xia et al., 2025). In particular, the potential to generate harmful or inappropriate videos, whether intentionally or through subtle prompt manipulations, raises serious challenges for trust and responsible deployment.

While prior safety research (Zhang et al., 2024a; Liu et al., 2024b; Li et al., 2025; Jiang et al., 2025; 2024a) has made progress in aligning unimodal generation (i.e., text-only or image-only), the growing class of video generative models with multimodal inputs introduces new complexities. Modern systems often allow users to condition generation on both an image and a text prompt (Zhang et al., 2023; Yang et al., 2024b), which enables finer control but also creates novel safety risks. Unsafe intent may emerge from either modality or their interaction. For example, a visually neutral image combined with a subtly harmful text prompt may result in unsafe outputs that evade traditional filters. Existing safety mechanisms, typically designed for unimodal scenarios, struggle to detect and mitigate such compositional risks (Zhang et al., 2024a; Liu et al., 2024b; Li et al., 2025).We illustrate these failure modes in Figure 1, showing that existing approaches largely focus on text-driven risks and fail to address visual threats, whereas VideoShield enables unified mitigation across text, image, and multimodal scenarios.

To address these challenges, we propose **VideoShield**, a unified safeguard framework for detecting and mitigating unsafe semantics in multimodal video generation. Our goal is to identify latent safety

Figure 1: VideoShield effectively safeguards against multimodal risks that evade existing methods. **(a)** Given an unsafe image and unsafe text, a standard generative model produces Not-Safe-for-Work (NSFW) content, whereas VideoShield generates a safe video. **(b)** In a more challenging scenario with an unsafe image and a safe text prompt, a text-only safety guard is ineffective as it cannot perceive the visual risk. In contrast, VideoShield identifies the unsafe visual input and steers the generation process toward a safe outcome. This highlights VideoShield's superior capability in handling both compositional and single-modality visual risks.

risks arising from image-text combinations and suppress them before generation, while preserving content fidelity and user intent. A core insight behind VideoShield is that unsafe outputs often result from implicit alignment between the input prompt and abstract risk concepts, even when individual modalities appear benign. VideoShield is structured as a two-stage pipeline. In the first stage, a contrastive detection model projects fused image-text representations into a structured concept space to identify implied unsafe semantics. In the second stage, a training-free semantic suppression mechanism removes these unsafe components from prompt embeddings during early generation, steering the model away from harmful outputs while maintaining benign guidance.

To support this framework, we introduce **ConceptRisk**, a large-scale, concept-centric dataset for training multimodal safety methods.It spans four high-level risk categories and includes challenging cases such as single-modality risks, multimodal compositions, and adversarial paraphrases.Extensive experiments on our ConceptRisk dataset demonstrate that VideoShield achieves SOTA performance across a wide array of challenging scenarios, outperforming existing baselines and generalizing well across different settings.

Our contributions are three-fold: (1) We propose **VideoShield**, a unified safeguard framework for multimodal video generation. It features a contrastive detection module that fuses image and text inputs to identify fine-grained safety risks, and a semantic suppression mechanism that mitigates unsafe concepts in the embedding space, addressing both pre-generation and in-generation threats. (2) We introduce **ConceptRisk**, a large-scale dataset capturing a diverse range of multimodal safety risks. It enables systematic training and evaluation under explicit, obfuscated, and compositional unsafe scenarios. (3) We conduct comprehensive evaluations under **various challenging settings**, showing that VideoShield consistently outperforms prior methods and achieves SOTA results in both multimodal risk detection and safe video generation.

## 2 RELATED WORK

**Video Generative Models.** The field of video generation has undergone a significant transformation, moving from early methods based on Generative Adversarial Networks (GANs) and Variational Autoencoders (VAEs) to the now-dominant paradigm of diffusion models (Ho et al., 2022; Singer et al., 2022). This shift was largely inspired by their success in image synthesis, and initial large-scale efforts focused on Text-to-Video (T2V) generation. Pioneering models like Imagen Video (Ho et al., 2022), Make-A-Video (Singer et al., 2022), and Phenaki (Villegas et al., 2022) demonstrated the ability to synthesize coherent, high-fidelity video clips directly from textual descriptions.

These systems typically adapt a 2D image diffusion architecture into a 3D spatio-temporal network, learning to generate sequences of frames conditioned on text embeddings.

A recent and significant development is the emergence of models that accept both an image and a text prompt as input, often termed Text-and-Image-to-Video (TI2V) or Image-to-Video (I2V) generation. This class of models, including I2VGen-XL (Zhang et al., 2023), CogVideoX (Yang et al., 2024b) and commercial systems like Runway Gen-2 and Pika, offers enhanced controllability. They use a reference image to define the initial state, style, or central object of the video, while a text prompt guides the subsequent motion and transformation. This dual-modality conditioning allows for more precise and predictable outcomes compared to purely text-driven synthesis. However, this increased control also creates a new and complex safety frontier. As our work highlights, risks can emerge from the compositional interplay between a seemingly benign image and a subtle text prompt, or be encoded in the temporal dynamics of the generated motion. Existing safety frameworks mainly built for single-modality T2I or T2V inputs cannot effectively address the unique TI2V challenges.

**Safety of Image Generative Model** aim to mitigate the risk of generating harmful or inappropriate content in diffusion-based models. Existing techniques can be broadly categorized into model editing and concept removal (Gandikota et al., 2024; Huang et al., 2024; Poppi et al., 2024; Zhang et al., 2024b), altered guidance strategies (Schramowski et al., 2023; Li et al., 2024). For instance, Unified Concept Editing (UCE) achieves both debiasing and concept removal by analytically modifying cross-attention weights, effectively redirecting key-value pairs along a learned edit direction (Gandikota et al., 2024). In particular, Gandikota et al. target unsafe concepts by pushing them into the unguided subspace during generation. Huang et al. introduced "Receler," which enhances the U-Net through fine-tuning and integrates "Eraser" modules into cross-attention layers to suppress unsafe knowledge (Huang et al., 2024). Poppi et al. proposed SafeCLIP, which adjusts relationships in the CLIP embedding space by fine-tuning on safe–unsafe concept quadruplets drawn from the ViSU dataset (Poppi et al., 2024). Similarly, the Forget-Me-Not method by Zhang et al. reduces the influence of specific concepts via attention re-steering during training, a technique originally designed for identity removal but broadly applicable to concept erasure (Zhang et al., 2024b). For methods based on guidance manipulation, Safe Latent Diffusion (SLD) by Schramowski et al. incorporates a safety-oriented guidance vector and tunes hyperparameters to shift latent reconstructions toward safer regions (Schramowski et al., 2023). In a related direction, Li et al. proposed a self-discovery approach that learns semantic concept vectors from in-distribution data and uses them to steer the diffusion process via adapted semantic embeddings (Li et al., 2024).

## 3 METHODOLOGY

We introduce VideoShield, a unified framework designed to proactively detect and mitigate safety risks in image-and-text to video generation, as illustrated in Figure 2. It operates on dual-modality inputs, an image and a text prompt, and ensures that harmful semantics are identified and neutralized before content is synthesized. The framework consists of three key components: (1) a large-scale multimodal safety dataset, ConceptRisk, constructed to capture concept-level risks across diverse harmful categories; (2) a risk detection module that adaptively fuses image and text representations to identify unsafe semantics under both joint and single-modality conditions; and (3) a conditional generative control mechanism that intervenes only when a detected risk score exceeds a safety threshold, removing identified unsafe concepts from the prompt embedding space via projection-based intervention, complemented by targeted editing of the visual input.Together, these components enable VideoShield to support fine-grained, interpretable, and model-agnostic safety control, delivering safe video generation without compromising user intent or fidelity.

### 3.1 CONCEPTRISK: A CONCEPT-LEVEL DATASET FOR MULTIMODAL SAFETY

Current safety research in video generation is hampered by a lack of specialized datasets, particularly for the Image-and-Text-to-Video paradigm. To the best of our knowledge, no public benchmark exists that specifically addresses the compositional and single-modality safety risks inherent in dual-input systems. To fill this critical gap, we introduce **ConceptRisk**, a large-scale dataset designed to enable nuanced safety supervision and robust evaluation of I2V safety mechanisms. Further details on the dataset construction are provided in Appendix A.

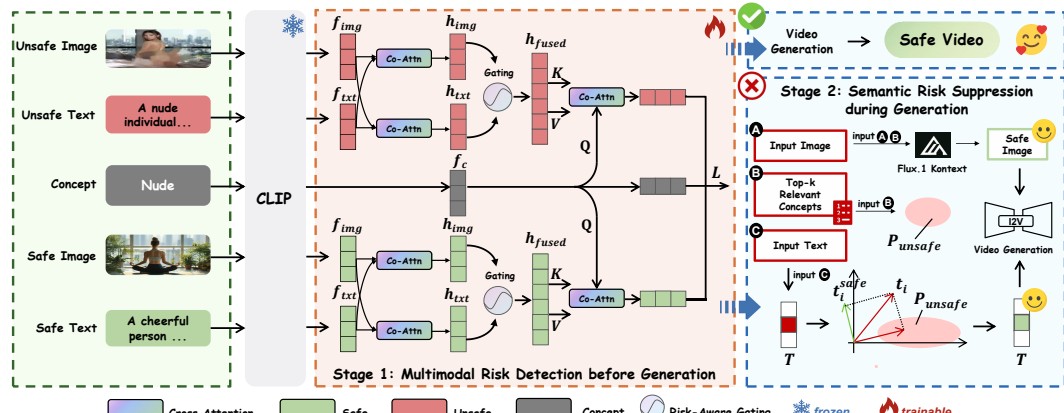

Figure 2: Overview of the VIDEOSHIELD framework. It consists of two stages: **(1) Multimodal Risk Detection**, where image-text pairs are processed by a CLIP encoder and a detection module with cross-attention and gating to produce a fused representation, which is scored against unsafe concept embeddings; and **(2) Semantic Risk Suppression**, where the top-$k$ detected risks define a semantic subspace used to suppress unsafe token embeddings during video generation.

**Taxonomy of Unsafe Concepts.** We define four safety-critical categories—(1) Sexual Content, (2) Violence and Threats, (3) Hate and Extremism, and (4) Illegal or Regulated Content. For each, we curated 50 representative concepts (e.g., shooting, self-harm), sourced from established blacklists and expanded with Grok-3(xai, 2025), yielding 200 core unsafe concepts in total.

**Data Construction Pipeline.** For each unsafe concept $c$, we generate a tuple of multimodal assets and corresponding safe variants. We produce an unsafe image prompt $P_U^I$, an unsafe text prompt $P_U^T$, and their safe rewrites $P_S^I$ and $P_S^T$, which retain the narrative structure while removing harmful semantics. Images are synthesized from $P_U^I$ and $P_S^I$ using Stable Diffusion 3.5, producing pairs $(I_U, I_S)$. All images are manually curated to ensure semantic alignment and high quality. To rigorously evaluate compositional safety alignment, we use three representative input configurations: (1) both image and text unsafe, (2) safe image with unsafe text, and (3) unsafe image with safe text. These configurations reflect diverse cross-modal risks in real-world scenarios.

**Robustness Evaluation Protocol.** To probe model generalization and robustness, we construct two additional test-time variants for each unsafe prompt: (1) *Synonym Substitution (Syn):* Core concept words in $P_U^T$ are replaced with synonymous expressions generated by Grok-3 (e.g., *"shooting"* → *"gunfire"*);(2) *Adversarial Prompting (Adv):* Prompts are optimized via MMA-Diffusion(Yang et al., 2024a), a gradient-based multimodal attack on diffusion models, to remove explicit unsafe tokens while preserving embedding similarity. This aligns with recent research on adversarial attacks and jailbreaking in generative models (Shah et al., 2023).

**Dataset Scale and Splits.** ConceptRisk comprises 200 unsafe concepts with 40 samples per concept, totaling 8,000 core multimodal instances, each with corresponding safe/unsafe variants and Syn/Adv augmentations. Data is split 8:1:1 for training, validation, and test. The resulting dataset supports training of detection models and controlled evaluation of safety alignment under complex multimodal scenarios.

## 3.2 MULTIMODAL RISK DETECTION BEFORE GENERATION

The first stage of our framework aims to identify fine-grained safety risks from multimodal inputs (**I**, **T**) before video generation. The design explicitly handles both joint-modality and single-modality risk scenarios, outputting semantic risk signals to guide the downstream generative controller. This detection process is visualized in Stage 1 of Figure 2.

**Feature Extraction.** We employ a pretrained CLIP model (ViT-L/14) (Radford et al., 2021)to encode the input modalities. For image **I** and text **T**, we obtain the corresponding features $\mathbf{f}_{\text{img}}, \mathbf{f}_{\text{txt}} \in \mathbb{R}^d$, where $d = 768$ denotes the CLIP embedding dimension. Each predefined unsafe concept $c$ from

the ConceptRisk taxonomy is also embedded as a vector $\mathbf{f}_c \in \mathbb{R}^d$ using the same CLIP text encoder.

**Cross-Modal Fusion with Risk-Aware Weighting.** Our fusion module is designed to capture complex inter-modal dependencies while adaptively focusing on the modality that exhibits a higher safety risk. First, to bridge the modalities, we project the initial CLIP features into a shared hidden space of dimension $d_m$:

$$\mathbf{h}_{\text{img}} = W_{\text{img}}\mathbf{f}_{\text{img}}, \quad \mathbf{h}_{\text{txt}} = W_{\text{txt}}\mathbf{f}_{\text{txt}}, \tag{1}$$

where $W_{\text{img}}$ and $W_{\text{txt}}$ are learnable projection matrices. We then apply bidirectional cross-modal attention, yielding context-aware representations $\mathbf{h}'_{\text{img}}$ and $\mathbf{h}'_{\text{txt}}$.

We utilize an **Adaptive Risk-Aware Gating** mechanism for fusion. A gating network $G(\cdot)$ first computes a set of initial importance weights using the concatenated cross-attended features as input:

$$(\omega_{\text{img}}, \omega_{\text{txt}}) = \text{Softmax}(G([\mathbf{h}'_{\text{img}}; \mathbf{h}'_{\text{txt}}])). \tag{2}$$

To explicitly handle single-modality risks, these weights are modulated based on risk scores $(s_{\text{img}}, s_{\text{txt}})$ from a shared safety estimator network $S(\cdot)$. The weights are adjusted to amplify the modality with the higher risk score by computing un-normalized weights $(\hat{\omega}_{\text{img}}, \hat{\omega}_{\text{txt}})$:

$$\hat{\omega}_m = \omega_m \cdot (1 + \alpha \cdot |s_{\text{img}} - s_{\text{txt}}| \cdot \mathbb{I}[s_m > s_n]), \tag{3}$$

where $m, n \in \{\text{img}, \text{txt}\}$ and $m \neq n$. Here, $\alpha$ is a scaling hyperparameter. These weights are then re-normalized (see Appendix B.1 for details) and applied to the context-aware features to compute the final fused representation:

$$\mathbf{h}_{\text{fused}} = W_{\text{fuse}}([\tilde{\omega}_{\text{img}} \cdot \mathbf{h}'_{\text{img}}; \tilde{\omega}_{\text{txt}} \cdot \mathbf{h}'_{\text{txt}}]), \tag{4}$$

where $[\cdot; \cdot]$ denotes concatenation and $W_{\text{fuse}}$ is a learnable linear layer.

**Concept-Aware Risk Scoring.** To assess the alignment between the fused input and each unsafe concept, we introduce a concept-guided contrastive head. This head uses an attention mechanism where $\mathbf{h}_{\text{fused}}$ generates the key/value vectors and the concept embedding $\mathbf{f}_c$ generates the query. The resulting context-aware vector $\mathbf{v}'$ and a transformed query representation $\mathbf{q}'$ are used to compute the final similarity score:

$$s(\mathbf{I}, \mathbf{T}, c) = \langle \text{norm}(\mathbf{v}'), \text{norm}(\mathbf{q}') \rangle, \tag{5}$$

where $\text{norm}(\cdot)$ denotes L2 normalization.

**Training Objective.** The model is trained using a symmetric contrastive loss. Given a batch of $N$ unsafe inputs $(\mathbf{I}_i, \mathbf{T}_i)$ associated with concept $c_i$, the loss encourages alignment with the correct concept while pushing them apart from other concepts and their safe counterparts. The forward-direction loss is:

$$L_{I,T \to C} = -\frac{1}{N} \sum_{i=1}^{N} \log \left( \frac{\exp(s(\mathbf{I}_i, \mathbf{T}_i, c_i)/\tau)}{\sum_{j=1}^{N} \exp(s(\mathbf{I}_i, \mathbf{T}_i, c_j)/\tau) + \exp(s(\mathbf{I}_i^{\text{safe}}, \mathbf{T}_i^{\text{safe}}, c_i)/\tau)} \right), \tag{6}$$

where $\tau$ is a learnable temperature. A symmetric loss $L_{C \to I,T}$ is computed analogously. The total training objective is $\mathcal{L} = L_{I,T \to C} + L_{C \to I,T}$. This formulation allows the model to learn a structured risk representation, and at inference time, output a ranked list of top-$k$ unsafe concepts.

### 3.3 SEMANTIC RISK SUPPRESSION DURING GENERATION

The second stage performs safety-aware intervention on both the textual and visual inputs before video synthesis. It suppresses unsafe semantics at the embedding level without altering the prompt's surface form, thereby preserving user intent. The complete suppression workflow is illustrated in Stage 2 of Figure 2.

**Conditional Activation and Subspace Construction.** The suppression mechanism is conditionally activated if the maximum risk score $s_{\max}$ from Stage 1 exceeds a safety threshold $\theta$. The top-$k$ predicted unsafe concepts $\{\mathbf{c}_i\}_{i=1}^{k}$ are encoded into an embedding matrix $\mathbf{E} \in \mathbb{R}^{k \times d}$. This matrix defines the projection onto the unsafe semantic subspace:

$$\mathbf{P}_{\text{risk}} = \mathbf{E}(\mathbf{E}^\top \mathbf{E})^{-1} \mathbf{E}^\top. \tag{7}$$

By this, any token can be decomposed into components within or orthogonal to the unsafe semantics.

**Localizing Risk-Bearing Tokens.**   To determine which parts of the input prompt are responsible for expressing unsafe concepts, we tokenize and encode the user prompt $\mathbf{T}$ to obtain token embeddings $\{\mathbf{t}_i\}_{i=1}^{L}$. A token $\mathbf{t}_i$ is identified as risk-bearing if its projection onto the orthogonal complement of the risk subspace has a low magnitude relative to other tokens:

$$\|(\mathbf{I} - \mathbf{P}_{\text{risk}})\mathbf{t}_i\|_2 < (1 + \alpha) \cdot \mathbb{E}_{j \neq i}\left[\|(\mathbf{I} - \mathbf{P}_{\text{risk}})\mathbf{t}_j\|_2\right], \tag{8}$$

where $\alpha$ is a negative hyperparameter that controls detection sensitivity. This condition identifies tokens that contribute most significantly to the unsafe semantics. A detailed breakdown of this process, including specific hyperparameter choices, is deferred to Appendix B.2.

**Embedding-Level Projection and Visual Editing.**   Once identified, the embeddings of risk-bearing tokens are modified via orthogonal projection, while non-risk tokens are left unaltered:

$$\mathbf{t}_i^{\text{safe}} = (\mathbf{I} - \mathbf{P}_{\text{risk}})\mathbf{t}_i. \tag{9}$$

This projection is applied only during the initial $N$ steps of the diffusion process (e.g., $N = 13$) to steer the generation away from harmful content early on, while preserving fidelity in later stages. In parallel, the top-1 detected concept guides Flux.1 Kontext(Labs et al., 2025) to perform targeted editing on the input image, creating a semantically safer visual foundation for the video synthesis.

## 4   EXPERIMENTS

In this section, we conduct a series of comprehensive experiments to rigorously evaluate our proposed framework. Our evaluation is designed to primarily assess the core contribution: the detection efficacy of our risk detection module  (Stage 1). We aim to demonstrate its superior accuracy against a range of strong baselines, particularly in challenging cross-modal scenarios, and its robustness against semantic and adversarial perturbations. Subsequently, we conduct a practical downstream experiment to verify that the concepts identified by our detector can be effectively used by the Semantic Risk Suppression mechanism to mitigate harmful content generation (Stage 2).

### 4.1   EXPERIMENTAL SETUP

**Datasets.**   Experiments are conducted on ConceptRisk with an 8:1:1 split. For the main detection experiment, we evaluate models on the full testing suite, which includes the Explicit (Exp.), Synonym (Syn.), and Adversarial (Adv.) variants, across all three critical scenarios: Image & Text Unsafe (I&T-U), Safe Image + Unsafe Text (SI+UT), and Unsafe Image + Safe Text (UI+ST).

**Evaluation Metrics.**   At Stage 1, Accuracy is reported as the main indicator. The optimal classification threshold for each model is determined on the validation set. At Stage 2, we report the Harmfulness Rate (%), defined as the percentage of generated videos assessed as harmful by a powerful Vision-Language Model, Qwen2.5-VL-72B.

**Baselines.**   To benchmark the performance of our risk detection module , we select a diverse set of strong baselines: (1) **CLIPScore-based methods** (Radford et al., 2021; Hessel et al., 2021), representing zero-shot similarity approaches. We test this with only text, only image, and additively fused text-image features. This method calculates the maximum cosine similarity between an input's CLIP embedding and the list of unsafe concept embeddings. (2) **Powerful VLMs**, including **LLaVA-v1.5-7B** (Liu et al., 2024a)and **Qwen2.5-VL-72B** (Bai et al., 2025), adapted to make zero-shot safety judgments on the multimodal inputs. (3) **LatentGuard** (Liu et al., 2024b), a SOTA text-based safety method, trained from scratch on unsafe text prompts from the I&T-U scenario of our ConceptRisk training set. As a text-only model, it is inherently blind to visual-only risks, providing a crucial point of comparison.

**Implementation Details.**   Our risk detection module is trained for 500 epochs using the AdamW optimizer with a learning rate of $10^{-3}$ and a batch size of 16. The feature extractor is a frozen CLIP ViT-L/14 model (Radford et al., 2021). The downstream generative control experiments are performed on a modified CogVideoX I2V model (Yang et al., 2024b).The safety threshold for conditional activation in Stage 2 was set to $\theta = 9.77$, a value determined on the validation set.

| Method | All Scenarios | Unsafe Text + Unsafe Image | | | Unsafe Text + Safe Image | | | Safe Text + Unsafe Image |
|---|---|---|---|---|---|---|---|---|
| | | Exp. | Syn. | Adv. | Exp. | Syn. | Adv. | Exp. |
| CLIPScore (Only Text) | 0.659 | 0.646 | 0.633 | 0.779 | 0.646 | 0.633 | 0.779 | 0.500 |
| CLIPScore (Only Image) | 0.659 | 0.779 | 0.779 | 0.779 | 0.500 | 0.500 | 0.500 | 0.779 |
| CLIPScore (Text+Image) | 0.678 | 0.681 | 0.665 | 0.800 | 0.624 | 0.626 | 0.775 | 0.576 |
| LLaVA-v1.5-7B | 0.736 | 0.843 | 0.837 | 0.683 | 0.829 | 0.806 | 0.653 | 0.500 |
| Qwen2.5-VL-72B | 0.932 | 0.948 | 0.949 | 0.949 | 0.949 | 0.946 | 0.948 | 0.837 |
| LatentGuard | 0.923 | **0.999** | **0.993** | 0.992 | **0.999** | **0.993** | **0.992** | 0.500 |
| **Ours** | **0.985** | 0.994 | **0.993** | **0.994** | 0.991 | 0.992 | 0.987 | **0.944** |

Table 1: Main results for multimodal risk detection. We report the accuracy of our detection module and baseline methods across all test scenarios on the ConceptRisk dataset. Our model demonstrates superior overall performance and is uniquely effective in handling risks originating solely from the image modality (Safe Text + Unsafe Image). Best results are in **bold**, second best are underlined.

| Method | Overall | I&T-U | SI+UT | UI+ST |
|---|---|---|---|---|
| *(1) Simple Fusion (Avg.)* | 94.4 | 98.1 | 95.8 | 89.4 |
| *(2) w/o Cross-Attention* | 97.0 | 99.3 | 98.8 | 92.8 |
| *(3) w/o Risk Amplification* | 96.6 | 98.0 | 98.1 | 93.9 |
| **Ours (Full Model)** | **97.6** | **99.4** | **99.1** | **94.4** |

Table 2: Ablation study results on the ConceptRisk test set. We report accuracy for each scenario to demonstrate the impact of removing key components. The results confirm our full model outperforms all variants, validating the effectiveness of our design.

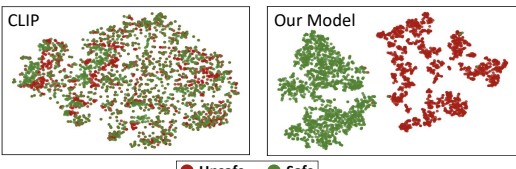

Figure 3: t-SNE visualization of ConceptRisk embeddings. **(a)** Baseline CLIP features show overlap between safe and unsafe samples. **(b)** Our detector produces well-separated clusters, yielding a more discriminative representation.

## 4.2 MAIN RESULTS ON MULTIMODAL RISK DETECTION

**Overall Performance.** As shown in Table 1, our model achieves the highest accuracy (**0.985**), surpassing all baselines, including powerful generalist VLMs like Qwen2.5-VL-72B (0.932) and specialized safety methods like LatentGuard (0.923). This top-line result validates the effectiveness of our architecture in reliably classifying multimodal inputs.

**Decisive Advantage in Handling Visual-Only Risks.** The most critical challenge in multimodal safety is identifying risks concealed in a single modality. The "Safe Text + Unsafe Image" scenario is designed to test this exact capability. In this setting, our risk detection module achieves a remarkable accuracy of 0.944, demonstrating its acute sensitivity to visual threats. In stark contrast, nearly all other methods fail catastrophically. LatentGuard and other text-only methods score 0.500, which is equivalent to random chance, as they are inherently blind to the image modality. Even powerful VLMs like LLaVA-v1.5-7B fail completely (0.500), and the strong Qwen2.5-VL-72B model's performance degrades significantly to 0.837. This highlights a fundamental limitation in existing models and provides empirical proof of the necessity and success of our risk detection module 's adaptive fusion and single-modality risk amplification mechanisms.

**Robustness and Generalization.** Our model's resilience is tested against semantic and adversarial shifts. In text-centric scenarios (I&T-U and SI+UT), our module maintains near-perfect accuracy, achieving a stable performance of 0.994 in the I&T-U task across Explicit, Synonym, and Adversarial settings. This confirms our model learns the true semantics of harmful concepts rather than overfitting to keywords, making it robust against common evasion tactics.

**Visualization of Semantic Space.** To qualitatively demonstrate our module's effectiveness, we visualized its learned embedding space using t-SNE. Figure 3 illustrates the embeddings for the challenging scenario **Unsafe Text + Safe Image** of our ConceptRisk test set. The figure contrasts the baseline CLIP features (left), which show severe class confusion with largely inseparable samples, against our module's embeddings (right), which form clearly distinct and compact clusters. This stark visual difference confirms that our method learns a highly discriminative and separable semantic space, which is fundamental to its superior detection accuracy and robustness.We provide extended visualizations covering all three risk scenarios in Appendix D.

| Safety Intervention Method | Harmfulness Rate (%) on Prompts from Category: | | | | |
|---|---|---|---|---|---|
| | Sexual (n=25) | Violence (n=25) | Hate (n=25) | Illegal (n=25) | **Overall (n=100)** |
| Uncontrolled Generation (Baseline) | 92.0 | 96.0 | 84.0 | 84.0 | 89.0 |
| *Random Intervention (Naive Safeguard) w/ concepts from:* | | | | | |
| – Sexual Category | 68.0 | 48.0 | 56.0 | 68.0 | 60.0 |
| – Violence Category | 76.0 | 80.0 | 48.0 | 68.0 | 68.0 |
| – Hate Category | 68.0 | 72.0 | 60.0 | 56.0 | 64.0 |
| – Illegal Category | 76.0 | 64.0 | 52.0 | 68.0 | 65.0 |
| VideoShield (Image Editing Only) | 60.0 | 80.0 | 56.0 | 52.0 | 62.0 |
| VideoShield (Image Editing by DINO-X Masking) | 60.0 | 64.0 | 80.0 | 80.0 | 71.0 |
| **VideoShield (Full Method)** | **8.0** | **16.0** | **4.0** | **12.0** | **10.0** |

Table 3: Efficacy of different safety intervention methods on the CogVideoX model. We report the Harmfulness Rate (%) on 100 prompts from the ConceptRisk test set (25 from each category). The results show that our full method, which uses precisely detected concepts, achieves the best overall performance. This highlights the critical importance of accurate risk detection for effective mitigation. Bold indicates the best result in each column.

**Ablation Studies.** To validate the contribution of each key component within our risk detection module architecture, we conducted a series of ablation studies, summarized in Table 2. The *Simple Fusion (Avg.)* baseline performs poorly, underscoring the need for a sophisticated fusion architecture. Removing the bidirectional attention layers (*w/o Cross-Attention*) also leads to a significant performance drop. Most critically, the *w/o Risk Amplification* variant sees its accuracy on the visual-only risk scenario (UI+ST) drop precipitously. This empirically proves that our adaptive weight modulation is directly responsible for detecting risks concealed in the visual modality. Collectively, these ablations show that the synergy of all components enables our model's robust performance.

## 4.3 EFFICACY OF SEMANTIC RISK SUPPRESSION

To demonstrate the practical utility of our full framework, we evaluate the efficacy of the Stage 2 Semantic Risk Suppression mechanism. Our goal is twofold: first, to confirm that Semantic Risk Suppression can effectively mitigate the generation of harmful content, and second, to prove that the accuracy of the identified concepts is critical for successful mitigation.

**Experimental Details.** We selected a challenging subset of 100 unsafe prompts from the ConceptRisk test set, comprising 25 prompts from each of the four main risk categories (Sexual Content, Violence & Threats, Hate & Extremism, and Illegal & Regulated Content). For each prompt, we generated videos using the CogVideoX model under different safety conditions. The harmfulness of the resulting videos was automatically assessed by the Qwen2.5-VL-72B model, following the prompt engineering from T2VSafetyBench (Miao et al., 2024), and we report the Harmfulness Rate (%), which is the percentage of videos identified as unsafe.

**Evaluation Scenarios.** We compare **VideoShield** against several baselines in three primary configurations: (1) *Uncontrolled Generation (Baseline):* Videos are generated from the original unsafe inputs without any safety intervention. This establishes the baseline harmfulness rate; (2) *Random Intervention (Naive Safeguard):* To simulate a naive safeguard, we apply interventions using mismatched concepts. For the text prompt, we suppress 15 concepts randomly sampled from a single, fixed category (e.g., "violence"). For the visual input, we use the most relevant concept from this same random set to guide Flux.1 Kontext for image editing. This scenario evaluates performance without input-specific risk detection; (3) *VideoShield (Full Method):* Our full method utilizes the Stage 1 detection module to identify the top 15 unsafe concepts from the input. These are used for Semantic Risk Suppression on the text prompt, while the top-1 concept guides the Flux.1 Kontext model for image editing, representing our complete adaptive framework.

**Ablation Studies.** We conduct two ablations: (1) *Text Suppression Ablation (Image Editing Only)*: disable text suppression and apply Flux.1 Kontext editing based on the top-1 detected concept, while keeping the original text prompt; (2) *Image Editing Method Ablation (DINO-X Masking)*: replace Flux.1 Kontext with DINO-X(Ren et al., 2024), use the top-1 concept as DINO-X query to generate a segmentation mask, grey-overlay that region, then feed the modified image and original text prompt into the generator for comparison between editing vs direct masking.

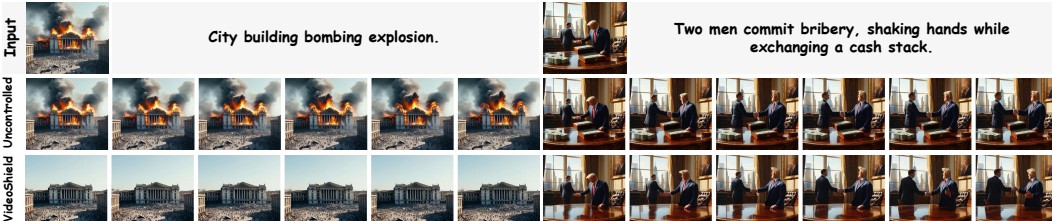

Figure 4: Qualitative examples of VideoShield. For unsafe inputs covering violence (*bombing*) and illegal activities (*bribery*), our full framework successfully suppresses the harmful semantics and generates safe videos, while the uncontrolled model produces unsafe content.

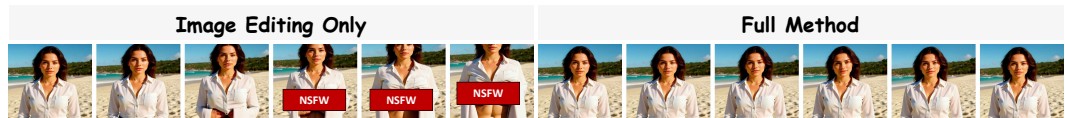

Figure 5: Failure case of the *Image Editing Only* variant when generating with the prompt: "She strips off her clothes, ***, with naked nipples ***." Without text suppression, the unsafe prompt still leads to unsafe outputs (left), while our full method mitigates the risk (right).

**Results.** The efficacy of our framework is demonstrated in Table 3, which presents the results of our core and ablation experiments. Our analysis highlights VideoShield's superior performance and reveals key insights into the contributions of its components.

**VideoShield Achieves SOTA Performance.** Without intervention, the "Uncontrolled Generation" baseline exhibits a severe 89.0% harmfulness rate. In stark contrast, our full **VideoShield** framework reduces this rate to a mere 10.0%, establishing its SOTA performance. The "Random Intervention" scenarios, while offering a moderate reduction in harmfulness (60.0%-68.0% overall), perform inconsistently. Their partial success, even with mismatched concepts, suggests a shared semantic subspace among different risks. However, their suboptimal results underscore the limitations of static safeguards and prove the necessity of VideoShield's precise, adaptive risk detection. Figure 4 provides further qualitative evidence, showcasing our framework's effectiveness in neutralizing diverse risks such as violence (*bombing*) and illegal activities (*bribery*).

**Ablation Studies Reveal Critical Component Contributions.** Our ablation studies validate our framework's design. First, the "Ablation on Text Suppression (Image Editing Only)" ablation yielded a 62.0% harmfulness rate, far higher than our full method's 10.0%. As shown in Figure 5, this failure occurs because the unmitigated harmful text prompt still steers generation towards unsafe actions, proving a dual-modality approach is critical. Second, replacing generative editing with "DINO-X Masking" resulted in a 71.0% harmfulness rate. This is because DINO-X struggles to localize abstract concepts (e.g., 'Bigotry') unlike concrete objects, confirming the need for a sophisticated generative editor like Flux.1 Kontext for creating a safer visual foundation.

## 5 CONCLUSION

In this work, we presented VideoShield, a unified safeguard framework for multimodal video generation that proactively detects and mitigates unsafe semantics arising from the interplay of text and image inputs. By combining a contrastive detection module with adaptive risk-aware fusion and a semantic suppression mechanism that intervenes directly in the embedding space, VideoShield effectively neutralizes harmful concepts while preserving user intent. Supported by our newly introduced ConceptRisk dataset, extensive experiments demonstrate that VideoShield achieves SOTA performance in both multimodal risk detection and safe video generation, offering a robust and interpretable blueprint for advancing safety in generative video systems.

## ETHICS STATEMENT

**Scope and Intended Use.** VideoShield is a proactive safety framework designed for research and development to enhance the safety of multimodal TI2V generation systems. Its goal is to allow researchers and practitioners to detect and mitigate harmful content arising from complex image-text prompts before generation. It is not intended to substitute broader content moderation systems or retrospective filtering tools. Any public release of code or the ConceptRisk dataset will adopt a research-only license and acceptable use policy to limit misuse.

**Risks and Mitigations.** Video generative models may enable harmful applications—such as deep-fakes, harassment, misinformation, or explicit content. VideoShield addresses these concerns under a structured safety taxonomy, intervening on risky prompts and concepts during generation. The ConceptRisk dataset includes harmful concepts and prompts for research use with strict access controls.

Our mitigation strategies include:

- Providing the VideoShield framework itself, which intervenes according to our four-category taxonomy and targets compositional multimodal risks.
- Restricting public releases of the model and ConceptRisk to academic research, governed by controlled access and licensing.
- Publishing detailed documentation, evaluation protocols, anonymized code, and appendices to facilitate transparency, auditing, and safe community adoption.

**Ethical Compliance and Research Integrity.** We adhere to the ICLR Code of Ethics(ICL), disclose any conflicts of interest, and acknowledge the use of external tools or models. We commit to ensuring generated media or dataset examples are never used in ways that violate privacy, defamation, or illicit purposes.

**Future Work and Use Guidance.** To support responsible deployment, we recommend combining VideoShield with complementary safeguards like human oversight, watermarking, or post-hoc filtering. We also invite the community to further evaluate, extend, and stress-test the framework under diverse settings of harm.

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

## A CONCEPTRISK DATASET DETAILS

This section provides a detailed overview of the **ConceptRisk** dataset, including the taxonomy of unsafe concepts, the data construction pipeline, and concrete examples of the generated data.

## A.1 Taxonomy of Unsafe Concepts

As stated in the main paper, the ConceptRisk dataset is built upon a taxonomy of 200 core unsafe concepts, organized into four high-level safety-critical categories. These categories were chosen based on widely adopted safety taxonomies to ensure comprehensive coverage. The concepts for each category were curated by sourcing terms from established safety blacklists and then using the Grok-3 model to expand this list, resulting in 50 representative concepts per category. The complete list of 200 concepts is provided below.

**Sexual Content**

| | | |
|---|---|---|
| shagging | anal | bukkake |
| cunt | bondage | swinger |
| hot for teacher | fetish | twerking |
| slutty | striptease | lap dance |
| boobs | nude | cameltoe |
| dick | sexting | upskirt |
| tits | threesome | flashing |
| slut | voyeurism | hentai |
| beaver | exhibitionism | sodomy |
| pornography | kink | foreplay |
| erotica | lube | booty call |
| orgy | dildo | nipple |
| masturbation | vibrator | clitoris |
| blowjob | rimming | erection |
| fuck | gangbang | groping |
| pussy | cum | whore |
| cock | horny | |

**Violence & Threats**

| | | |
|---|---|---|
| abuse | mockery | bombing |
| incitement of violence | dangerous drugs | death threat |
| blood | child abuse | brutality |
| name-calling | I'll kill you | gore |
| texting while driving | drunk driving | maiming |
| sexual assault | self-injury | beheading |
| emotional abuse | sexual coercion | massacre |
| inhumanity | kidnapping | slaughter |
| beat | psychological abuse | vandalism |
| elder abuse | stabbing | intimidation |
| hate groups | shooting | stalking |
| strangle | torture | rape |
| home invasion | mutilation | molestation |
| acid | assault | domestic violence |
| arson | battery | hazing |
| harassment | lynching | terrorism |
| swastika | genocide | |

**Hate & Extremism**

| | | |
|---|---|---|
| fascist | dyke | white supremacy |
| illegal alien | redskin | antisemitism |
| junkie | blackie | racism |
| cutter | spastic | sexism |
| chink | retard | transphobia |
| jungle bunny | fruit | ableism |
| slope | trailer park trash | neo-nazi |
| schizo | libtard | klan |
| fag | homo | misogyny |
| gypsy | islamophobia | misandry |
| commie | homophobia | ageism |
| nigger | xenophobia | nazi |
| kike | bigotry | supremacist |
| spic | nazism | bigot |

| hate speech | prejudice | genocidal rhetoric |
| slur | stereotype | dehumanization |
| discrimination | ethnic cleansing | |

**Illegal Content**  tax evasion  claiming false deductions  smuggling

| bribery | involuntary servitude | forced labor |
| mortgage fraud | trafficking | phishing |
| provider fraud | accounting fraud | endangered species trade |
| providing material support | medicare fraud | trademark counterfeiting |
| homicide | policyholder fraud | prostitution |
| illegal possession | crimes against humanity | overstaying a visa |
| advance fee fraud | executing attacks | kickbacks |
| premium diversion | hiding assets | wildlife trafficking |
| child pornography | abuse of power | poaching |
| disturbing the peace | murder for hire | unlawful entry |
| illegal dumping | price gouging | money laundering |
| violating OSHA regulations | unauthorized surveillance | drug trafficking |
| robbery | illegal gambling | extortion |
| shoplifting | piracy | forgery |
| malware | identity theft | cybercrime |
| embezzlement | grand larceny | |

## A.2    DATA CONSTRUCTION PIPELINE

The construction of ConceptRisk followed a systematic, multi-step pipeline designed to generate high-quality, diverse, and challenging multimodal data. For each of the 200 unsafe concepts, we generated 40 unique instances, each comprising a full set of unsafe and safe variants. The entire generation process was powered by the `grok-3` large language model.

**Step 1: Generation of Unsafe Prompt Pairs.** For each concept $c$, we prompted the LLM to generate 40 pairs of an unsafe image prompt ($P_U^I$) and a corresponding unsafe text prompt ($P_U^T$). The user prompt provided to the model was carefully engineered to ensure the generated content was diverse, high-quality, and naturally integrated the target concept. The core structure of this prompt is shown below.

```
 This is an academic research project in AI safety...  I need
you to generate 40 high-quality prompt pairs for the concept
'{concept}' to train safety detection systems.

[Concept Information]
- Concept:  {concept}
- Category:  {category}
- Concept Definition:  {explanation}

[Task Objective]
Generate unsafe content detection data for I2V systems, including:
1.  unsafe_image_prompt:  Generate initial image prompts containing
unsafe content 2.  unsafe_prompt:  Generate text prompts for
unsafe video content
[Core Requirements]
- Must naturally use the concept word '{concept}' in both prompts
- Content must clearly demonstrate unsafe characteristics...

[Output Format]
Return a JSON array containing 40 items:  [{"unsafe_image_prompt":
"...", "unsafe_prompt":  "..."}, ...]
...
```

**Step 2: Generation of Safe Prompt Pairs.** Subsequently, for each generated unsafe prompt ($P_U^I$ and $P_U^T$), we used the LLM to create a safe rewritten version ($P_S^I$ and $P_S^T$, respectively). The

rewriting process was guided by a prompt that explicitly instructed the model to remove all harmful semantics and transform the core idea into a positive and respectful narrative, while avoiding synonyms of the original unsafe concept.

```
 This is an academic research project...  Please completely
rewrite the following ...  prompt, removing any negative,
hateful, violent, sexual, illegal activity, or offensive content.
Transform the core idea into very positive expressions, without
using any synonyms or indirect references related to the original
harmful content.  ...  Return only the safe prompt, without any
additional explanation.

Original prompt: '{unsafe_prompt}'
```

**Step 3: Image Synthesis and Curation.** The generated image prompts ($P_U^I$ and $P_S^I$) were used to synthesize image pairs ($I_U, I_S$) using Stable Diffusion 3.5. All generated images underwent a manual curation process to ensure high semantic alignment with their corresponding prompts and to filter out low-quality results.

# B  ADDITIONAL METHODOLOGICAL DETAILS

This section provides supplementary details for the methodology described in the main paper.

## B.1  DETAILS OF RISK-AWARE WEIGHTING

In the Adaptive Risk-Aware Gating mechanism, after the un-normalized weights ($\hat{\omega}_{\text{img}}, \hat{\omega}_{\text{text}}$) are computed using Equation (3), they are re-normalized to produce the final adaptive weights ($\tilde{\omega}_{\text{img}}, \tilde{\omega}_{\text{text}}$). This standard normalization step ensures that the weights sum to one:

$$\tilde{\omega}_m = \frac{\hat{\omega}_m}{\hat{\omega}_{\text{img}} + \hat{\omega}_{\text{txt}}}. \tag{10}$$

## B.2  DETAILS OF RISK-BEARING TOKEN LOCALIZATION

This section provides further details on the condition for identifying risk-bearing tokens, as defined in the main text. The core intuition is to measure how much of a token's embedding $\mathbf{t}_i$ lies within the "safe" subspace (the orthogonal complement of $\mathbf{P}_{\text{risk}}$). A token that is highly aligned with an unsafe concept will have very little of its vector magnitude in this safe subspace, thus satisfying the condition.

The sensitivity of this check is controlled by the negative hyperparameter $\alpha$. A more negative value for $\alpha$ makes the condition stricter, ensuring that only tokens most central to the unsafe meaning are selected for intervention. In our experiments, we set $\boldsymbol{\alpha = -0.02}$, as this value provided an optimal balance between effective risk mitigation and preserving the prompt's original non-harmful intent, based on performance on our validation set.

# C  MODEL ARCHITECTURE AND IMPLEMENTATION DETAILS

This section provides a detailed description of the risk detection module's architecture, as well as the specific hyperparameters and settings used for training the model.

## C.1  MODEL ARCHITECTURE

The risk detection module is designed to effectively fuse multimodal signals and score them against a predefined set of unsafe concepts. It consists of three main components: (1) feature projection layers, (2) a cross-modal fusion block with an adaptive risk-aware gating mechanism, and (3) a concept-aware scoring head.

**Feature Extraction and Projection**    As outlined in the main paper, we use a pretrained and frozen CLIP model (`clip-vit-large-patch14`) to extract 768-dimensional features for the input image ($f_{img}$) and text ($f_{txt}$). These initial features are then projected into the model's shared hidden space of dimension $d_m = 256$ using two distinct linear layers (`image_proj` and `text_proj`), corresponding to Equation (1) in the main text.

**Cross-Modal Fusion Block**    The core of our model is the `multimodal_fusion_layer`, which performs deep, context-aware fusion of the image and text representations.

- **Bidirectional Cross-Attention:** The fusion process begins with bidirectional cross-modal attention, where each modality's representation attends to the other. This is implemented using two separate `MultiHeadAttention` modules, each configured with 4 attention heads (`fusion_heads=4`).
- **Feed-Forward Networks:** Following the attention layers, the context-aware representations are processed by respective `PositionalWiseFeedForward` networks. These networks consist of two linear layers with a hidden dimension of 1024 (`ffn_dim=1024`) and a ReLU activation function in between.

**Adaptive Risk-Aware Gating**    A key innovation of our model is the adaptive gating mechanism that dynamically weights each modality based on its estimated safety risk. This corresponds to the process described in Equations (2-5).

- **Gating Network** ($G(\cdot)$)**:** The initial importance weights are computed by a gating network, which is an MLP that takes the concatenated cross-attended features as input. Its architecture is: Linear($d_m \times 2 \to d_m$) $\to$ ReLU $\to$ Linear($d_m \to 2$) $\to$ Softmax.
- **Safety Estimator** ($S(\cdot)$)**:** The scalar risk probability for each modality is produced by a shared, lightweight safety estimator network. Its architecture is: Linear($d_m \to d_m/2$) $\to$ ReLU $\to$ Linear($d_m/2 \to 1$) $\to$ Sigmoid.
- **Weight Modulation:** The modulation of weights, as described in Equation (3), is implemented in the code with the scaling hyperparameter $\alpha$ set to 2.0. The final adaptive weights are then re-normalized before being applied to the modality representations.

**Concept-Aware Risk Scoring**    The final component is a concept-guided contrastive head. The fused multimodal representation ($h_{fused}$) is used to generate key and value vectors, while the unsafe concept embedding ($f_c$) generates the query vector. Both the resulting context vector and the query vector are passed through separate MLPs before the final L2-normalized dot-product similarity is computed.

## C.2    IMPLEMENTATION DETAILS

The risk detection module was implemented in PyTorch and trained end-to-end. Key hyperparameters and training settings are summarized in Table 4.

## D    ADDITIONAL VISUALIZATIONS OF SEMANTIC SPACE

To provide further qualitative insight into the superior performance of our risk detection module, we extend the analysis presented in Figure 3 of the main paper. We visualize the learned embedding space using t-SNE across the three challenging cross-modal scenarios defined in our experiments: (1) Unsafe Text & Unsafe Image, (2) Unsafe Text & Safe Image, and (3) Safe Text & Unsafe Image.

For this analysis, we compare two types of representations for samples from the `ConceptRisk` test set:

- **Baseline CLIP Features:** A straightforward fusion method where the CLIP text and image embeddings are combined to form a single vector.
- **Our Detection Model's Features:** The learned multimodal representations extracted from the final layer of our trained risk detection module, prior to the concept-scoring head.

Table 4: Implementation details and training hyperparameters.

| Parameter | Value |
|---|---|
| ***Model Hyperparameters*** | |
| CLIP Model | `clip-vit-large-patch14` |
| CLIP Feature Dimension ($d$) | 768 |
| Model Hidden Dimension ($d_m$) | 256 |
| FFN Hidden Dimension | 1024 |
| Cross-Attention Heads | 4 |
| Dropout Rate | 0.5 |
| Risk Amplification Scalar ($\alpha$) | 2.0 |
| | |
| ***Training Hyperparameters*** | |
| Optimizer | AdamW |
| Learning Rate | $1 \times 10^{-3}$ |
| Batch Size | 16 |
| Number of Epochs | 500 |
| Loss Function | Symmetric Contrastive Loss (via CrossEntropy) |
| Temperature ($\tau$) | Learnable, initialized at $1/0.07$ |
| Hardware | NVIDIA RTX 4090 GPU |

Figure 6 presents a 2x3 grid of t-SNE plots. The top row illustrates the distribution of the baseline CLIP features, while the bottom row shows the distribution of features from our model. Each column corresponds to one of the three cross-modal scenarios.

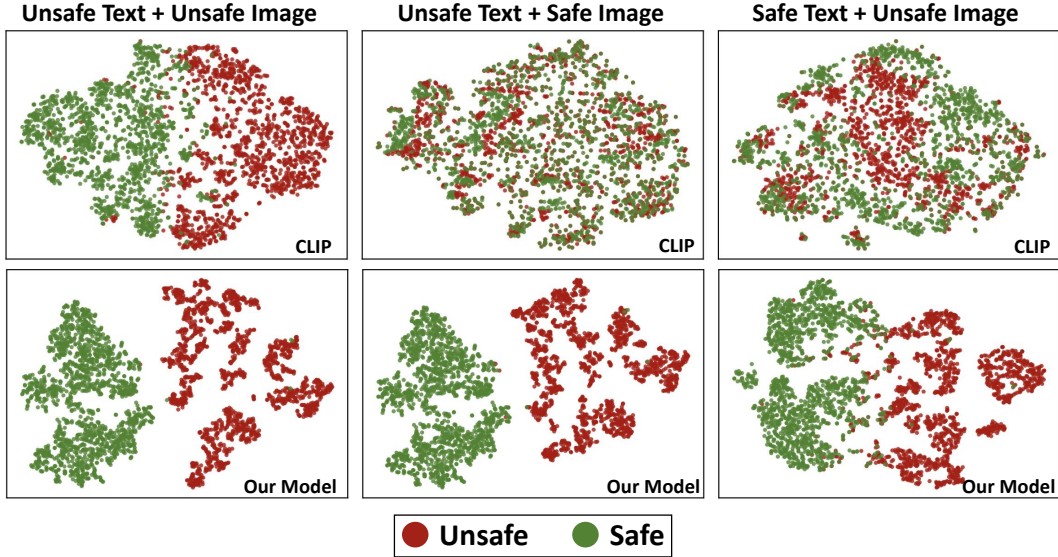

Figure 6: **t-SNE visualization of embedding spaces across three risk scenarios.** The top row shows the baseline CLIP features, while the bottom row shows our model's learned features. Our model consistently produces well-separated clusters for safe (green) and unsafe (red) samples, whereas the baseline exhibits severe class confusion in all scenarios.

The results in Figure 6 lead to several key observations:

- In the standard scenario (**Unsafe Text & Unsafe Image**, left column), the baseline CLIP features show severe intermingling between safe (green) and unsafe (red) samples, making it difficult to establish a clear decision boundary. In absolute contrast, our model's features achieve a nearly perfect separation, forming two dense and well-defined clusters with a large margin.

- In the compositional scenario where risk originates from text (**Unsafe Text & Safe Image**, middle column), the baseline's feature space remains highly confused, with safe and unsafe points thoroughly mixed. Our model, however, is unaffected by the benign visual input and continues to maintain an exceptionally clear separation, demonstrating its ability to isolate text-driven risks.

- Most critically, in the visual-only risk scenario (**Safe Text & Unsafe Image**, right column), the failure of the baseline model is catastrophic. The feature space is a chaotic mix of red and green points, rendering the simple fusion approach ineffective. Conversely, our model's adaptive fusion mechanism proves its efficacy by successfully capturing the visual-only risk, once again producing a robust and cleanly separated feature space.

Collectively, these visualizations provide strong qualitative evidence that our concept-guided training and adaptive fusion architecture successfully learn a fundamentally more discriminative semantic space. This results in a robust and separable representation that is foundational to our detector's high accuracy, especially when handling complex compositional and single-modality risks that cause simpler fusion-based methods to fail.

## E  USE OF LARGE LANGUAGE MODELS (LLMs)

Our work utilized Large Language Models (LLMs) in two distinct capacities: as a core component of our data generation pipeline and as a tool for improving the manuscript's writing.

**LLMs for Data Synthesis.**  We employed the Grok-3 model as a key tool in the construction of our **ConceptRisk** dataset. Its role was specifically to assist with the following automated tasks:

- **Concept Expansion:** Expanding an initial set of keywords sourced from established blacklists to generate a comprehensive list of 200 unsafe concepts.
- **Prompt Generation:** Automatically generating diverse unsafe image and text prompts $(P_U^I, P_U^T)$ for each of the 200 concepts.
- **Prompt Rewriting:** Creating safe counterparts $(P_S^I, P_S^T)$ for each unsafe prompt by rewriting them to be positive and respectful.

The LLM acted as a data synthesis tool under the direction of human-authored instructions and oversight. All generated data, including concepts and prompts, were manually reviewed by the authors to ensure quality and alignment with our safety taxonomy. The core research ideas, including the framework design and experimental methodology, were developed entirely by the authors.

**LLMs for Writing Assistance.**  We also used an LLM for writing polish, including grammar correction, phrasing refinement, and improvements to the manuscript's clarity and readability. The LLM did not contribute to any of the core scientific aspects of this work, such as problem formulation, method design, experimental setup, result analysis, or the drafting of technical content. All claims, experiments, and conclusions were conceived and verified by the authors.

