# OpenReview forum: "VideoShield: A Unified Framework for Multimodal Risk Detection and Control in Video Generative Models"
_ICLR.cc/2026/Conference — ICLR 2026 Conference Withdrawn Submission_

### Official Review · Reviewer_LPpw · 2025-10-23

**Soundness:** 2
**Presentation:** 3
**Contribution:** 3
**Rating:** 4
**Confidence:** 3

**Summary:**

This paper proposes VideoShield, a unified safeguard framework to proactively detect and mitigate unsafe semantics in multimodal video generation, solving the new safety risks emerged from video generative models which fail to be defended by existing safety methods. Furthermore, they introduce ConceptRisk, a large-scale, concept-centric dataset to captures various multimodal safety scenarios. Comprehensive experiments have been studied to support the superiority of VideoShield.

**Strengths:**

1.	The proposed two-stage pipeline is novel and effectively address safety risks across modality of both image and text modal.

2.	Constructing a novel ConceptRisk dataset with unsafe concepts to train a defense framework is well-motivated and effective.

3.	Detection performance seem good on ConceptRisk dataset compared with existing baselines.

**Weaknesses:**

1.	The proposed evaluation only focuses on ConceptRisk dataset. As your VideoShield is trained on this dataset, this will introduce potential overfit to this safety/harmful distribution, and it is unfair for baseline aligned model like LLaVA-V and Qwen-VL.

2.	Furthermore, deployment of VideoShield to real-world text-to-video models seems difficult. To fit VideoShield, you need to have both text prompts and relative images. This may be not appliable for modern T2V models as no relative image holds. A potential solution could raise from generate input image based on the prompt from some T2I models, but this induces large computational overheads.

3.	Evaluation on semantic risk suppression is only based on CogVideoX model. Lack of results on other T2V models like Pika, Lumiere, Sora, etc.

**Questions:**

1.	How does your defense framework generalize to real-world scenarios as no input images included?

2.	Can you provide computational cost of training VideoShield based on ConceptRisk dataset, and the inference cost compared with existing methods like LatentGuard?

3.	Can you evaluate the possibility of over-refusal when using VideoShield? I think VideoShield may result in significant degradation of model utility as lack of such evaluation in the paper.

I will adjust my rating if authors can address my concerns well, especially about real-world application and potential over-refusal of VideoShield.

---

### Official Review · Reviewer_Mrgs · 2025-10-28

**Soundness:** 2
**Presentation:** 2
**Contribution:** 1
**Rating:** 2
**Confidence:** 4

**Summary:**

The paper
- Constructs a fully synthetic text-image dataset based on 200 keywords (‘concepts’) from 4 safety categories. The dataset contains basic combinations of unsafe text and images and safe re-writes of the unsafe data.
- Employs a contrastive loss to train a joint text, image embedding on top of a pre-trained clip model to embed samples close to the relevant keyword and far away from other keywords and safe text, image embeddings.
- Uses the resulting embedding space for a) unsafe content detection on a held-out set of the constructed data and b) to sanitize text, image inputs to a video generation model.

**Strengths:**

- The counterfactual data synthesis with pairs of safe and unsafe data is a good strategy to isolate unsafe features in the dataset.
- The contrastive loss that enables fine-grained safety classification with respect to the keywords makes sense

**Weaknesses:**

- The work completely ignores the large amount of existing literature and datasets for vision-language safety, e.g. [1-5].
- No demonstration of the method generalizability beyond their dataset.
- The dataset seems completely solvable by a combination of image-only and text-only classifiers. The LatentGuard baseline solves all unsafe text cases and I imagine a simple clip-based classifier finetuned on their unsafe images would be able to perform on par with the proposed method.
- So overall the advantage of the proposed method for vision language safety classification over existing methods is not demonstrated.
- The paper motivates the method to safeguard video generation, but nothing in the method is video specific and no reasonable baselines are proposed. E.g. one could use a off-the-shelf vision language model to rewrite the text input and to propose a safety edit for the image before passing it to the video generation
- The video generation eval only measures harmfulness but no quality metric e.g. of how closely the generated video resembles the original user query

[1] Liu, Xin, et al. "Mm-safetybench: A benchmark for safety evaluation of multimodal large language models." European Conference on Computer Vision. Cham: Springer Nature Switzerland, 2024.

[2] Hu, Xuhao, et al. "Vlsbench: Unveiling visual leakage in multimodal safety." arXiv preprint arXiv:2411.19939 (2024).

[3] Wang, Siyin, et al. "Safe Inputs but Unsafe Output: Benchmarking Cross-modality Safety Alignment of Large Vision-Language Model." arXiv preprint arXiv:2406.15279 (2024).

[4] Röttger, Paul, et al. "MSTS: A Multimodal Safety Test Suite for Vision-Language Models." arXiv preprint arXiv:2501.10057 (2025).

[5] Lee, Youngwan, et al. "HoliSafe: Holistic Safety Benchmarking and Modeling with Safety Meta Token for Vision-Language Model." arXiv preprint arXiv:2506.04704 (2025).

**Questions:**

- Several parts in the architecture are named with high-level concepts, e.g. “safety estimator network” or “Risk-Aware Weighting”. What is the evidence that these architecture parts are actually performing those functions?
- Minor:
    - The paper should have a content warning in the beginning
    - Figure 1) is not fully censored, parts of breasts and nipples are still visible.

---

### Official Review · Reviewer_mnfM · 2025-10-31

**Soundness:** 2
**Presentation:** 3
**Contribution:** 2
**Rating:** 4
**Confidence:** 5

**Summary:**

This paper presents VideoShield, a framework for risk detection and control in multimodal video generation models. It operates in two stages: first, a risk detection module identifies potential hazards by combining image and text inputs, and second, a semantic suppression mechanism removes harmful concepts during the generation process. To support this framework, the authors introduce the ConceptRisk dataset, which is designed for training and evaluating multimodal safety methods. Experiments show that VideoShield outperforms existing approaches in both risk detection and safe video generation.

**Strengths:**

1. The paper explores the safety risks in video generation models caused by multimodal inputs, a factor that previous works have not fully considered. It offers a new perspective on video safety.
2. The paper is clearly written.

**Weaknesses:**

1. There is a lack of direct comparison with key baselines. The baselines mainly include simple CLIP-based methods and large multimodal models, with the only defense method being LatentGuard. This doesn't fully demonstrate the superiority of VideoShield. It should also compare with more defense methods, such as SafeWatch or others [1][2]. Additionally, the experimental results should be included in Table 1 in the main text.

[1] SafeWatch: An Efficient Safety-Policy Following Video Guardrail Model with Transparent Explanations

[2] Safree: Training-free and adaptive guard for safe text-to-image and video generation

2. There is a lack of results on other key datasets. Evaluating VideoShield only on the ConceptRisk dataset proposed in the paper is insufficient and does not adequately demonstrate its effectiveness. Moreover, since ConceptRisk is used both for training and testing, it fails to show VideoShield's generalization across different data sources. It should also be evaluated on other datasets, such as T2VSafetyBench [3] (since T2VSafetyBench appears to be text-only, it should be paired with safe or unsafe images). The experimental results should also be added to Table 1 in the main text to objectively validate VideoShield’s effectiveness.

[3] T2vsafetybench: Evaluating the safety of text-to-video generative models

3. Multimodal safety defense mechanisms have already been applied. For example, the open-source Stable Diffusion uses a text-image-based safety filter that blocks generated images when the cosine similarity between the CLIP embedding of the generated image and any pre-computed CLIP text embeddings of 17 unsafe concepts exceeds a threshold. This contradicts what is stated in the introduction of the paper.

I will adjust the score based on the author's responses and improvements to these issues.

**Questions:**

na

---

### Official Review · Reviewer_2bNv · 2025-11-10

**Soundness:** 2
**Presentation:** 1
**Contribution:** 1
**Rating:** 2
**Confidence:** 5

**Summary:**

This paper proposes to tackle the task of "generation with safety (i.e. removing unsafe concepts behind the prompt for proper generation)" with the particular focus on multimodal settings (i.e. the input prompt is multimodal, while the unsafe concepts could exist either in textual or visual prompts, or even both), which is claimed to be the first of its kind. The proposed method, named VideoShield, is a unified safeguard framework, which has a contrastive detection module that fuses image and text inputs to identify fine-grained safety risks, and a semantic suppression mechanism that mitigates unsafe concepts in the embedding space. In order to conduct the model training and experiments, a large-scale dataset (named as ConceptRisk) capturing a diverse range of multimodal safety risks is collected, in which different scenarios are supported (i.e. unsafe text+unsafe image, unsafe text+safe image, safe text+unsafe image).

**Strengths:**

+ The proposed ConceptRisk dataset seems to be the first of its kind, which specifically addresses the compositional and single-modality safety risks inherent in dual-input systems. Moreover, such dataset is composed of samples from diverse unsafe concepts.
+ Moreover, the evaluation protocol contains the verification upon the robustness, in which synonym substitution and adversarial prompting are adopted (where the latter aligns with the recent research of adversarial attacks and jailbreaking in generative models).
+ The method learns the joint embedding space which not only fuses the features from different input modalities but also aligns with the unsafe concept embeddings, such joint embedding is beneficial for identifying/detecting the unsafe concepts in the input prompts.

**Weaknesses:**

- The proposed method seems to lack for the novelty. In particular, the techniques used for performing semantic risk suppression are from the existing ones: 1) for moving the unsafe tokens from the textual embedding, the proposed method is highly similar to the work from Yoon et al., SAFREE: Training-Free and Adaptive Guard for Safe Text-to-Image And Video Generation, ICLR 2025, in which this work is not even cited; 2) while for the visual embedding, the technique of Flux.1 Kontext from Labs et al., 2025 is directly utilized.
- Moreover, while the semantic risk compression in the proposed method is based on two existing approaches, the suppressions between textual and visual embedding are not well aligned, i.e. the suppression upon textual embedding is based on the top-k predicted unsafe concepts while the one upon visual embedding is only applied for the top-1 detected concept. And there is no proper discussion upon such alignment and its potential issue for the further generation.
- The description of the proposed has some essential details missing, e.g. the safety estimator network for predicting the risk scores are not properly explained (we don't even know what is the input for such safety estimator network).
- The ablation study does not present significant differences among various design choices (i.e. the degradation of model variants compared to the proposed method is actually marginal) hence being unconvincing to verify the contribution of proposed design choices.
- The evaluation seems to lack for the proper/fair baseline. For instance, the most competitive baseline being used for experiments is LatentGuard, which is a text-only detection model thus the comparison is actually unfair. Instead, there should be a naive baseline of directly combining LatentGuard and image-based detector (e.g. image-only CLIPScore). Moreover, the comparison against the VLMs might be also unfair, as both LLaVA and Qwen are zero-shot while the proposed method not only tunes to detect the unsafe concepts but also adopts the validation set to discover the optimal threshold. We also note that, even Qwen is a zero-shot detector, its performance is already getting close to the proposed method (0.932 versus 0.985 for all scenarios), the improvement made by the proposed method hence becomes insignificant.

**Questions:**

The authors should carefully address the aforementioned weaknesses (i.e. unclear description for the proposed method, limited novelty and missing reference, the potential concern upon unaligned concept removals across textual and visual embeddings, insignificant ablation study, the issue upon unfair comparison and missing baseline) in the rebuttal to overturn the current negative rating.

---

### Note · Authors · 2025-11-12

I have read and agree with the venue's withdrawal policy on behalf of myself and my co-authors.